# Hypercapnia as a Double-Edged Modulator of Innate Immunity and Alveolar Epithelial Repair: A PRISMA-ScR Scoping Review

**DOI:** 10.3390/ijms26199622

**Published:** 2025-10-02

**Authors:** Elber Osorio-Rodríguez, José Correa-Guerrero, Dairo Rodelo-Barrios, María Bonilla-Llanos, Carlos Rebolledo-Maldonado, Jhonny Patiño-Patiño, Jesús Viera-Torres, Mariana Arias-Gómez, María Gracia-Ordoñez, Diego González-Betancur, Yassid Nuñez-Beyeh, Gustavo Solano-Sopó, Carmelo Dueñas-Castell

**Affiliations:** 1Group of Intensive Care and Comprehensive Care (GRIMICI), Barranquilla 080002, Colombia; dairorodelo1992@gmail.com (D.R.-B.); marianav-ariasg@unilibre.edu.co (M.A.-G.); mariacler11@hotmail.com (M.G.-O.); dagobuto@gmail.com (D.G.-B.); dryassid@hotmail.com (Y.N.-B.); 2Department of Intensive Medicine, Clínica Iberoamérica, Barranquilla 080002, Colombia; carlos.rebolledo@unisimon.edu.co (C.R.-M.); jesusviera9@gmail.com (J.V.-T.); 3Group Care Medicine, Clínica Colsanitas, Bogotá D.C. 110931, Colombia; 4Department of Internal Medicine, University of Cartagena, Cartagena de Indias130005, Colombia; josegabriel2101@gmail.com (J.C.-G.); jjpp0097@gmail.com (J.P.-P.); 5Department of Critical Medicine and Intensive Care, Faculty of Medicine, Simón Bolívar University, Barranquilla 080002, Colombia; 6Department of Anesthesiology and Perioperative Medicine, Faculty of Medicine, University of Sabana, Chía 250001, Colombia; camibonillallanos@gmail.com; 7Department of Critical Medicine, Clínica Gestión Salud, Santa Marta 470001, Colombia; gustavoa.solano1@gmail.com; 8Intensive Care and Obstetrics Research Group (GRICIO), University of Cartagena, Cartagena de Indias 130005, Colombia; crdc2001@gmail.com

**Keywords:** hypercapnia, PaCO_2_, innate immunity, alveolar epithelial repair, scoping review

## Abstract

Lung-protective ventilation and other experimental conditions raise arterial carbon dioxide tension (PaCO_2_) and alter pH. Short-term benefits are reported in non-infectious settings, whereas infection and/or prolonged exposure are typically harmful. This scoping review systematically maps immune-mediated effects of hypercapnia on innate immunity and alveolar epithelial repair. Scoping review per Levac et al. and PRISMA Extension for Scoping Reviews (Open Science Framework protocol: 10.17605/OSF.IO/WV85T; post hoc). We searched original preclinical studies (in vivo/in vitro) in PubMed, Web of Science, ScienceDirect, Cochrane Reviews, and SciELO (2008–2023). PaCO_2_ (mmHg) was prioritized; %Fraction of inspired Carbon Dioxide (%FiCO_2_) was recorded when PaCO_2_ was unavailable; pH was classified as buffered/unbuffered. Data were organized by context, PaCO_2_, and exposure duration; synthesis used heat maps (0–120 h) and a narrative description for >120 h. Mechanistic axes extracted the following: NF-κB (canonical/non-canonical), Bcl-2/Bcl-xL–Beclin-1/autophagy, AMPK/PKA/CaMKKβ/ERK1/2 and ENaC/Na,K-ATPase trafficking, Wnt/β-catenin in AT2 cells, and miR-183/IDH2/ATP. Thirty-five studies met the inclusion criteria. In non-infectious models, a “protective window” emerged, with moderate PaCO_2_ and brief exposure (65–95 mmHg; ≤4–6 h), featuring NF-κB attenuation and preserved epithelial ion transport. In infectious models and/or with prolonged exposure or higher PaCO_2_, harmful signals predominated: reduced phagocytosis/autophagy (Bcl-2/Bcl-xL–Beclin-1 axis), AMPK/PKA/ERK1/2-mediated internalization of ENaC/Na,K-ATPase, depressed β-catenin signaling in AT2 cells, impaired alveolar fluid clearance, and increased bacterial burden. Chronic exposures (>120 h) reinforced injury. Hypercapnia is a context-, dose-, time-, and pH-dependent double-edged modulator. The safe window is narrow; standardized, parallel reporting of PaCO_2_ and pH—with explicit comparisons of buffered vs. unbuffered hypercapnia—is essential to guide clinical translation.

## 1. Introduction

Acute lung injury (ALI), clinically represented by acute respiratory distress syndrome (ARDS), remains associated with high mortality and substantial global health-system burden [1,2,3]. Lung-protective ventilation strategies have reduced ventilator-induced lung injury (VILI) and ARDS mortality [4,5,6]; however, they often lead to alveolar accumulation of carbon dioxide (CO_2_) and a compensatory fall in pH over prolonged periods [4,5]. This condition—permissive hypercapnia—has been considered useful in selected ARDS scenarios and in obstructive lung diseases [7,8,9]. Yet recent studies suggest that hypercapnia may act not only as a severity marker but also as an independent predictor of mortality in specific clinical contexts [10,11,12,13]. Clinical evidence remains heterogeneous, and mechanistic uncertainties persist regarding how elevated CO_2_ affects lung tissue and host responses [14,15].

In preclinical models, hypercapnia modulates key immune and epithelial axes: it attenuates or reprograms NF-κB signaling (canonical and non-canonical) [16,17] and activates CaMKKβ/AMPK/PKA/ERK1/2 cascades that promote endocytosis of epithelial ion transporters (ENaC and Na,K-ATPase), with a direct impact on alveolar fluid clearance (AFC) [18,19]. The direction of effect depends on the biological context (infectious vs. non-infectious), dose (arterial carbon dioxide tension [PaCO_2_]), exposure duration, and acid–base status (buffered vs. unbuffered hypercapnia) [20]. Thus, a “double-edged” profile emerges: a potentially protective window with brief exposures and moderate PaCO_2_ [20] versus a detrimental profile—functional immunosuppression and epithelial dysfunction—when exposure is prolonged, PaCO_2_ is high, or active infection is present [21,22].

Beyond ARDS, chronic conditions such as chronic obstructive pulmonary disease (COPD), status asthmaticus, and cystic fibrosis can evolve to acute hypercapnic respiratory failure [23], particularly during infectious exacerbations, and are linked to poor outcomes [21,24]. These observations suggest that elevated CO_2_ may act as a causal modulator of innate immunity and epithelial homeostasis—rather than a mere epiphenomenon [23,25].

Despite the clinical and biological relevance, no integrative synthesis has specifically centered on the immune-mediated effects of hypercapnia on the pulmonary epithelium while standardizing exposure metrics (PaCO_2_ versus %FiCO_2_) and pH. To address this gap, we conducted a PRISMA-ScR–conformant scoping review that maps experimental evidence published between 2008 and 2023, stratified by an infectious/non-infectious context, PaCO_2_, exposure duration, and pH buffering. Our objective was to identify and characterize how hypercapnia modulates innate immunity and alveolar epithelial repair in experimental models. Specifically, we asked the following: How do these effects vary by context (infectious vs. non-infectious), PaCO_2_, and exposure time?

## 2. Results

The electronic search identified 3843 records, plus 116 from other sources (total 3959). After de-duplication (EndNote X8), 2576 unique references remained. Title/abstract screening excluded 2435 records. We assessed 141 full texts and excluded 106 that did not meet the research question. In total, 35 studies met the inclusion criteria (Figure 1). General characteristics (author, year, country, and design) are summarized in Table 1.

### 2.1. Dose- and Time-Dependent Effects by Context

#### 2.1.1. Non-Infectious Models

In non-infectious models (VILI/sterile mechanical stress), the median PaCO_2_ was 90.3 mmHg (IQR: 68.4–120), and median exposure was 4 h (IQR: 1–10.5) [16,17,18,19,20,26,27,29,31,32,34,35,36,37,38,39,41,42,43,44,45,47,48,49,50]. Within this setting, a potentially protective window emerged with moderate PaCO_2_ (~65–95 mmHg) and brief exposures (≤4–6 h), associated with NF-κB attenuation and epithelial preservation [20,32,36,37,41]. Conversely, a harmful window appeared at PaCO_2_ ≥ 110–120 mmHg, even with minutes-to-hours exposures, consistent with ENaC/Na,K-ATPase endocytosis and reduced AFC [18,26,38,42,47,48]. See Figure 2A.

#### 2.1.2. Infectious Models

In infectious models, the median PaCO_2_ was 69.8 mmHg (IQR: 60–78.3) and median exposure was 6 h (IQR: 5.5–25.5) [10,21,22,23,25,28,30,33,40,46]. A short favorable window (≈4–6 h) was observed in some models [28,30,40]. However, with longer duration (≥24–96 h) and/or higher PaCO_2_, harmful effects predominated: higher bacterial burden, impaired phagocytosis, and worse physiological outcomes [21,22,25]. Detrimental signals were more consistent under unbuffered conditions and without antibiotics [21,23,25]; with antibiotics and/or buffered pH, the signal often attenuated or became mixed [28,30], with notable exceptions even under buffering [22]. See Figure 2B.

Chronic exposures (>120 h) reinforced the harmful window in a hypercapnia-level-dependent manner [21,47,49].

### 2.2. Convergent Mechanisms

#### 2.2.1. NF-κB Signaling (Canonical and Non-Canonical), Stress-Kinase Signaling (ASK1/JNK/p38), and Innate Immunity Outputs (Cytokines, Phagocytosis, Autophagy)

Transcriptional alterations in innate immunity were reported in 34.3% (12/35) of studies [16,17,20,27,29,31,33,36,39,40,41,50]. Canonical NF-κB attenuation was the most frequent mechanism (6/12) [20,33,39,40,41,50]. Innate immune impairment (reduced phagocytosis and poorer microbial control) was documented in 51.4% (18/35) [10,17,21,22,23,25,27,28,29,30,31,32,33,36,37,39,43,46], especially in infectious in vivo models [10,21,23]. One cell study showed reduced autophagy and bacterial killing under hypercapnia [25]. While the global signal suggests immunosuppression with infection and/or prolonged exposure [17,23,32,33,36,37,39,43,46], some results were variable, reflecting model heterogeneity [10,28,31,50]. In sterile settings, early, moderate hypercapnia (PaCO_2_ 80–100 mmHg; ≤4 h) inhibits ASK1 and downstream JNK/p38 [39]. See Table 2.

#### 2.2.2. cAMP/PKA–AMPK Pathways and Epithelial Transport (ENaC; Na,K-ATPase; PKC-ζ; CaMKKβ)

Alveolar epithelial disruption/resealing with hypercapnia was observed in 37.1% (13/35) [18,19,22,26,27,34,35,42,44,45,47,48,50]. Multiple studies show that elevated CO_2_ promotes Na,K-ATPase (and ENaC) endocytosis, reduces membrane density, and impairs AFC [18,26,35,38,45,47,48]. Additional findings highlight the effects on epithelial repair, including suppressed Wnt/β-catenin in AT2 and mitochondrial/miR-183 signals [34,49]. In an infectious in vivo model with hypercapnic ventilation, alveolar transudation, septal edema, and greater alveolar damage were reported [27]. See Table 2.

## 3. Discussion

Hypercapnia is defined as an elevation of PaCO_2_ beyond the physiological range [51]. It occurs in chronic pulmonary diseases [52] and as a consequence of lung-protective ventilation in ARDS [4,5]. Multiple studies link hypercapnia to worse outcomes in the critically ill [12,13,53,54]. Our synthesis maps, across preclinical models, a context–dose–time–pH-dependent “double-edged” effect: in non-infectious settings, moderate PaCO_2_ with brief exposures associates with protective signals (attenuated inflammatory pathways and epithelial preservation); whereas in infectious models and/or with prolonged duration or high PaCO_2_, detrimental signals predominate (oxidative stress, impaired microbial control, epithelial dysfunction, and reduced AFC). See Figure 3.

### 3.1. Non-Infectious Models

In the absence of pathogens (VILI/sterile mechanical stress), the picture is nuanced and suggests a potentially protective window around PaCO_2_ ~65–95 mmHg with brief exposure (≤4–6 h). Within this range, several studies report canonical NF-κB attenuation, decreased IL-6/CXCL2, and epithelial preservation with improved microvascular leak/oxidation markers [32,36,41]. These effects track with ASK1–JNK/p38 inhibition, caspase-3 modulation, and reduced oxidative damage [32,36]. Still, signals are not uniform: PaCO_2_ 55–65 mmHg for 4 h has been associated with increased IL-8, VCAM-1, E-selectin/P-selectin, and macrophage alterations [36]; at 24 h, epithelial chemokines (CXCL1/2/6, CCL28, CXCL14) shift [43]. Nitrotyrosine elevations have also been reported despite anti-inflammatory signals [32,36]. Overall—even without infection—directionality depends on dose–time, species/cell line, sterile stimulus, and pH. In the heat maps (Figure 2A), the protective sector clusters at short exposure + moderate PaCO_2_, whereas higher thresholds or longer durations tip toward harm. Mechanistic integration in Figure 4A,B.

### 3.2. Infectious Models

Sepsis (pneumonia/systemic infection) is a major cause of severe ALI [55]. In this context, hypercapnia remodels host–pathogen interactions via intracellular pathways that affect pro-inflammatory cytokines (TNF and IL-6), phagocytosis, and autophagy [23,25,28]. Although a narrow favorable window (~4–6 h) is described in some models, the overall effect in the presence of infection is predominantly detrimental as exposure lengthens (≥24–96 h) or PaCO_2_ rises. This pattern—consistent with the 0–120 h heat maps (Figure 2) and reinforced by chronic exposures (>120 h)—presents as increased bacterial burden, depressed neutrophil/macrophage phagocytosis, autophagy inhibition (increased Bcl-2/Bcl-xL binding to Beclin-1), and deterioration of lung architecture [21,23,25,28]. Acid–base status is a key modulator: effects are seen with both buffered and unbuffered hypercapnia, but magnitude and even direction can differ [22,28]. We therefore separated these conditions and prioritized PaCO_2_ over inspired %CO_2_ whenever possible. In untreated infection, hypercapnia fosters bacterial propagation/replication, impairs host defense, and aggravates ALI—arguing for caution with prolonged/intense CO_2_ exposure in infectious settings [21,23,25]. Mechanistic integration in Figure 4A,B.

The safety of hypercapnia in pulmonary sepsis is a critical question in ICU patients [28]. Early hypercapnic acidosis within the first 24 h of mechanical ventilation has been associated with higher mortality in ARDS [54], alongside increased lung stiffness and airway pressures [56]. In severe pneumonia, “permissive hypercapnia” did not reduce mortality and was linked to greater morbidity [57]. Rigorous clinical trials are needed to define indications and safety thresholds.

### 3.3. Immunologic Effects of Hypercapnia

The NF-κB family provides an explanatory scaffold for lung injury, inflammation, and repair [41,58]. Across multiple models, CO_2_ ≥ 10% for ≥60 min inhibits the canonical pathway by reducing IKK complex phosphorylation, preserving IκBα, and preventing p65 nuclear translocation—dampening pro-inflammatory transcription [12,20,33,39,40,41,50,51,59,60]. In parallel, the non-canonical arm can be engaged via p100 processing to p52 and RelB nuclear localization, producing an immunosuppressive tone that persists for hours [16,17]. After stimulus withdrawal, many changes reverse within minutes, underscoring exposure-time dependence [16,17,33,40,50]. Context-dependent observations (PP2A–p65 axis with pro-inflammatory readout) emphasize modulation by pH, cell type, and microenvironment [16,17,18,19,20,26,27,29,31,32,34,35,36,37,38,39,41,42,43,44,45,47,48,49,50], contrasting PP2A’s classic negative regulation of NF-κB and suggesting key roles in intracellular homeostasis [61]. These axes are summarized in Figure 4A.

### 3.4. Alveolar Epithelial Repair/Healing

A second critical layer is ion-transporter trafficking and epithelial homeostasis. Hypercapnia activates CaMKKβ→AMPK and, in parallel, increases cAMP via adenylyl cyclase (AC)—especially with buffered pH—activating PKA (PKA-Iα). Together with PKC-ζ and ERK1/2, these routes converge to promote Na,K-ATPase and ENaC endocytosis, reduce their basolateral membrane density, and impair AFC [18,26,35,38,42,45,47,48,62,63,64]. Additional effects include assembly defects of Na,K-ATPase and suppression of Wnt/β-catenin in AT2 cells with reduced proliferation/repair after injury [49]. Notably, CO_2_-triggered AC rapidly elevates scAMP (≈15–30 min), phosphorylating PKA-Iα and initiating Na,K-ATPase internalization; this axis runs in parallel and complements the CaMKKβ→AMPK route [19,34]. These effects can appear from 30 min at PaCO_2_ ≥ 60 mmHg, intensify with longer exposures, and have been described with or without acidosis [18,26,42,45]. Figure 4C–E synthesizes these routes: AMPK/PKA and ENaC/Na,K-ATPase endocytosis; β-catenin/AT2; AC/cAMP/PKA-Iα; and miR-183/IDH2/ATP.

### 3.5. Limitations

This scoping review offers a comprehensive view of hypercapnia’s effects on innate immunity and the alveolar epithelium. Several limitations merit consideration: (i) heterogeneity in models and exposure metrics (PaCO_2_ vs. %CO_2_) precluded meta-analysis and motivated a direction-of-effect synthesis; (ii) OSF registration post hoc, dual screening, decision log, and sensitivity analyses mitigate—but do not eliminate—bias; (iii) potential publication bias (time window, databases, language); and (iv) limited clinical translation from in vivo/in vitro models. Accordingly, our patterns should be interpreted as operational hypotheses. Even so, these data provide a basis for studies that define thresholds and mechanisms with clinical relevance.

### 3.6. Clinical Implications and Future Directions

Taken together with the visual synthesis (heat maps, Figure 2; mechanistic schema, Figure 4), hypercapnia should be understood as a double-edged modulator whose effect depends on context (infectious vs. non-infectious), dose (PaCO_2_), exposure time, and pH. In non-infectious settings, we identify a potential protective window at moderate PaCO_2_ (~65–95 mmHg) and brief exposure (≤4–6 h), where canonical NF-κB inactivation, cytokine reduction, and preservation of epithelial ion transport predominate [10,28,30,39]. In contrast, with active infection and/or prolonged exposure (≥24–96 h) or high PaCO_2_ (≥110–120 mmHg), the signal shifts toward harm: autophagy inhibition (Bcl-2/Bcl-xL–Beclin-1 axis), depressed phagocytosis, AMPK/PKA/ERK1/2-mediated ENaC/Na,K-ATPase endocytosis, and reduced β-catenin in AT2 cells, with worse AFC and higher bacterial burden [19,25,34,49,50]. pH modulation (buffered vs. unbuffered) contributes to heterogeneity, reinforcing the need to co-report PaCO_2_ and pH [27,37].

Clinically, these observations favor a prudent, titrated approach. When hypercapnia arises from lung-protective ventilation, tolerance should be time-limited and constrained to a “protective window” of PaCO_2_, with close monitoring of pH, airway pressures, oxygenation, and infectious context. In uncontrolled infection or when long exposures/high PaCO_2_ are anticipated, risk–benefit appears unfavorable: Figure 4’s axes predict functional immunosuppression (impaired autophagy/phagocytosis) and epithelial dysfunction (altered ion trafficking and depressed AT2 repair) that may facilitate bacterial proliferation and dissemination. Permissive hypercapnia should therefore be avoided or minimized in untreated infection and, when unavoidable, restricted to brief exposure with appropriate antimicrobials and explicit PaCO_2_/pH targets. 

From a translational standpoint, we propose a PaCO_2_–time framework, modulated by pH and context, to standardize experimental design and clinical hypotheses: (i) systematically report PaCO_2_ (mmHg), pH, and exposure time; (ii) explicitly compare buffered vs. unbuffered hypercapnia and their interaction with AMPK/PKA and Bcl-2/Bcl-xL–Beclin-1 axes; (iii) define reproducible dose–time curves to bound protective windows and damage thresholds; (iv) incorporate biomarkers of epithelial dysfunction (membrane density of ENaC/Na,K-ATPase; β-catenin/AT2 markers) and innate immunity (phagocytosis and autophagy flux); and (v) evaluate therapeutic synchrony with antibiotics and pH-buffering strategies. Together, Figure 2 and Figure 4 offer an operational, biologically plausible framework to guide prudent clinical decisions and to design studies that confirm—or refute—the existence of a narrow safety window for hypercapnia.

### 3.7. The “Double-Edged Sword” of Hypercapnia: A Context-Specific Therapeutic Framework

Before context-specific recommendations, we set two operational principles: (i) CO_2_ effects depend on dose (PaCO_2_), exposure time, and pH, shifting the NF-κB–Ca^2+^/CaMKKβ–AMPK/PKA–β-catenin network between transient anti-inflammatory dampening and immuno-epithelial failure; (ii) clinical decisions are not binary but a titrated, reversible tolerance, guided by biomarkers (AFC, membrane ENaC/Na,K-ATPase density, β-catenin/AT2, phagocytosis/autophagy flux). With these principles, we distinguish the following.

#### 3.7.1. Non-Infectious Context (VILI/Sterile Stress)

Consider time-limited tolerance to moderate PaCO_2_ (≈65–95 mmHg) for ≤4–6 h, with pH closely monitored/preferably buffered, aiming to dampen canonical NF-κB without triggering epithelial failure loops.

#### 3.7.2. Infectious Context (Pneumonia/Sepsis)

Avoid or minimize hypercapnia—especially prolonged or high (≥110–120 mmHg)—due to functional immunosuppression (autophagy/phagocytosis) and barrier failure (ENaC/Na,K-ATPase). If unavoidable, limit duration, set explicit PaCO_2_/pH targets, and synchronize with antibiotics and source control.

## 4. Materials and Methods

This scoping review followed the PRISMA-ScR guidance (Preferred Reporting Items for Systematic Reviews and Meta-Analyses—extension for scoping reviews) and methodological recommendations by Levac et al. [65]. The protocol was publicly registered on the Open Science Framework (OSF) on 30 August 2024 (https://doi.org/10.17605/OSF.IO/WV85T). Because registration occurred after defining the study period, it is considered post hoc; any subsequent adjustments were documented in a decision log (Appendix A). The PRISMA-ScR checklist is provided in the Appendix A.

### 4.1. Eligibility Criteria

We included original preclinical studies (in vivo/in vitro/ex vivo) evaluating CO_2_ exposure/intervention and pulmonary immune and/or epithelial outcomes. Given the limited human clinical evidence, we considered experimental animal models and relevant cell lines/cultures (with or without acute lung injury). The preferred exposure metric was PaCO_2_ (mmHg); when only % inspired CO_2_ was reported, it was recorded descriptively. We distinguished between buffered and unbuffered hypercapnia. Outcomes covered the following: (i) innate immunity (phagocytosis, cytokines, canonical/non-canonical NF-κB); (ii) stress-kinase pathways (ASK1/JNK/p38); and (iii) CaMKKβ/AMPK/PKA/ERK1/2 axes linked to epithelial transport (ENaC, Na,K-ATPase, fluid clearance/integrity). We included studies published from Jan-2008 to Dec-2023 in English or Spanish. We excluded reviews, communications without primary data, clinical studies without an experimental component, models without documented epithelial impact, or lacking minimal CO_2_/pH reporting.

### 4.2. Information Sources and Search Strategy

We implemented a two-pronged search strategy. Electronic searches were run in PubMed, Web of Science, ScienceDirect, Cochrane Reviews, and SciELO. We combined MeSH terms and keywords, including “acute respiratory distress syndrome,” “pneumonia,” “ventilator-associated pneumonia,” “hypercapnia,” and “anti-inflammatory.” The full strategy is reported in Table A1. To broaden coverage, we performed manual reference screening of the included articles. Duplicates were managed in EndNote X8.

### 4.3. Study Selection

Title/abstract screening and full-text assessment were performed independently and in duplicate by two reviewers; discrepancies were resolved by consensus or a third reviewer. We did not compute κ coefficients, as the aim was to map the evidence (PRISMA-ScR scoping review); we prioritized a systematic, documented resolution of disagreements. We did not conduct a formal risk-of-bias assessment (SYRCLE) given the descriptive nature of the synthesis; relevant design domains were extracted and reported in a structured manner.

### 4.4. Data Extraction

Data were extracted using standardized templates (double entry with cross-checks). Variables included the following: model type; species/strain or cell line; context (infectious/non-infectious); acid–base status (buffered/unbuffered); exposure metric (PaCO_2_ in mmHg; %CO_2_ if only metric available); exposure duration (h); immune outcomes (NF-κB, cytokines, phagocytosis, autophagy); epithelial outcomes (ENaC, Na,K-ATPase, AFC/integrity); mechanistic axes (canonical/non-canonical NF-κB; ASK1/JNK/p38; CaMKKβ/AMPK/PKA/ERK1/2; AC/cAMP/PKA-Iα; miR-183/IDH2; Wnt/β-catenin in AT2 cells); and direction of effect (protective/mixed-neutral/harmful). Effect classification followed a pre-specified operational definition based on primary physiological/biological outcomes.

### 4.5. Evidence Synthesis and Mapping

Extracted data were tabulated for a comprehensive descriptive synthesis aligned with the review’s objective and research question. For the acute window (0–120 h), we built heat maps stratified by context (infectious/non-infectious), with PaCO_2_ and time discretization and effect coding (+1 = protective; 0 = mixed/neutral; −1 = harmful). When multiple data points were mapped to one cell, we averaged the score; missing cells were left blank. Chronic exposures (>120 h) were described separately and not integrated into the acute maps. Additionally, we created mechanistic panels for canonical/non-canonical NF-κB, ASK1/JNK/p38, and CaMKKβ/AMPK/PKA/ERK1/2 with epithelial transport (ENaC/Na,K-ATPase/AFC). Descriptive statistics (medians [p25–p75]) were produced in Jamovi 2.2.5. Operational details (PaCO_2_/time discretization, coding rules, sensitivity analyses) are provided in the Appendix A.

### 4.6. Transparency and Data Availability

We deposited in OSF the anonymized dataset, extraction templates, the decision log (Appendix A), the PRISMA-ScR checklist, and the full search strategies (Table A1).

## 5. Conclusions

This scoping review, organized within a PaCO_2_–time–pH framework and stratified by context (infectious vs. non-infectious), synthesizes the effects of hypercapnia on innate immunity and alveolar epithelial integrity/repair. Taken together, the evidence supports a narrow, context-dependent window in which moderate PaCO_2_ and brief exposure are associated with inflammatory attenuation and preservation of epithelial ion transport. Outside this window—particularly with active infection, prolonged exposure, or high PaCO_2_—a detrimental phenotype predominates, characterized by functional immunosuppression (autophagy blockade) and epithelial dysfunction (endocytosis of ENaC and Na,K-ATPase with depressed AT2-mediated repair), with potential adverse prognostic implications. These findings support hypercapnia as a dose-, time-, and pH-dependent double-edged modulator and underscore the need to co-report PaCO_2_ (mmHg), pH, and exposure time and to stratify by infection. The integrative framework presented here provides operational criteria to guide prudent clinical decision-making and to design translational studies that validate—or refute—the existence of a safe window for hypercapnia and delineate its damage thresholds with precision.

## Figures and Tables

**Figure 1 ijms-26-09622-f001:**
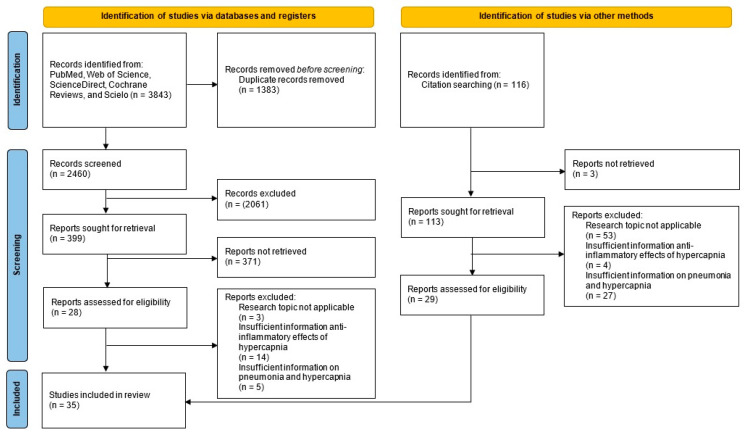
PRISMA-ScR flow diagram.

**Figure 2 ijms-26-09622-f002:**
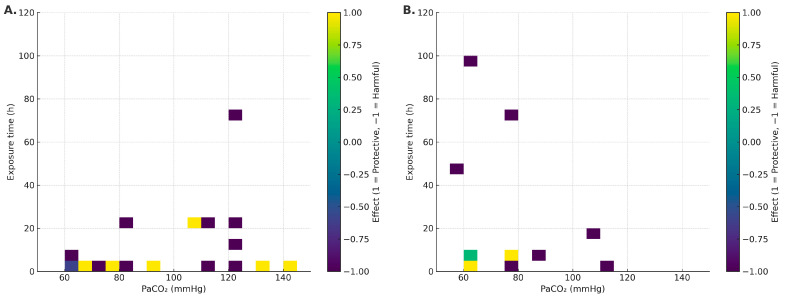
Heat maps of effect direction (−1 harmful, 0 mixed/neutral, +1 protective) within the acute window (0–120 h), stratified by context: (**A**) non-infectious; (**B**) infectious. Axes: PaCO_2_ (mmHg) and duration (h).

**Figure 3 ijms-26-09622-f003:**
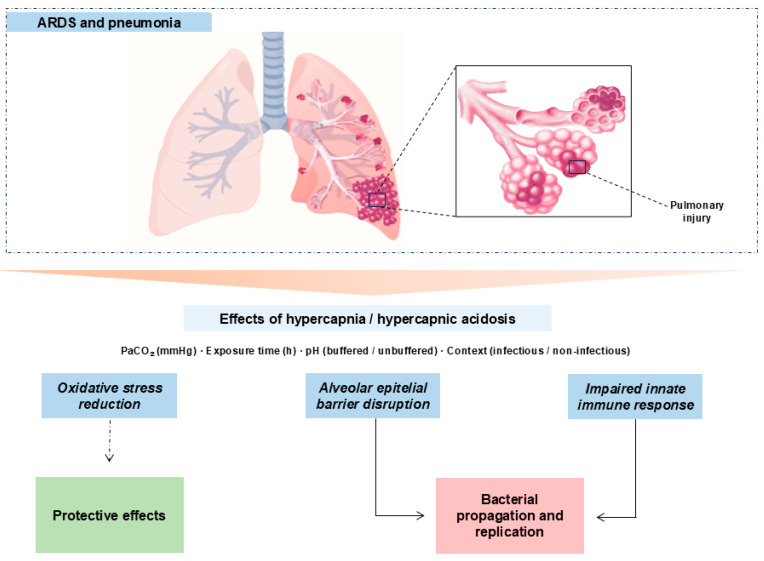
Mechanistic schema of hypercapnia/hypercapnic acidosis in experimental lung injury. Hypercapnia, modulated by PaCO_2_, exposure time, pH (buffered vs. unbuffered), and context (infectious vs. non-infectious), can transiently dampen inflammatory tone (potentially protective) but also disrupt the alveolar epithelial barrier and compromise innate immunity. In infection and/or with prolonged exposures or high PaCO_2_, detrimental pathways dominate with bacterial proliferation and dissemination. Solid arrows: primary relationships; dashed arrows: context-contingent protective links.

**Figure 4 ijms-26-09622-f004:**
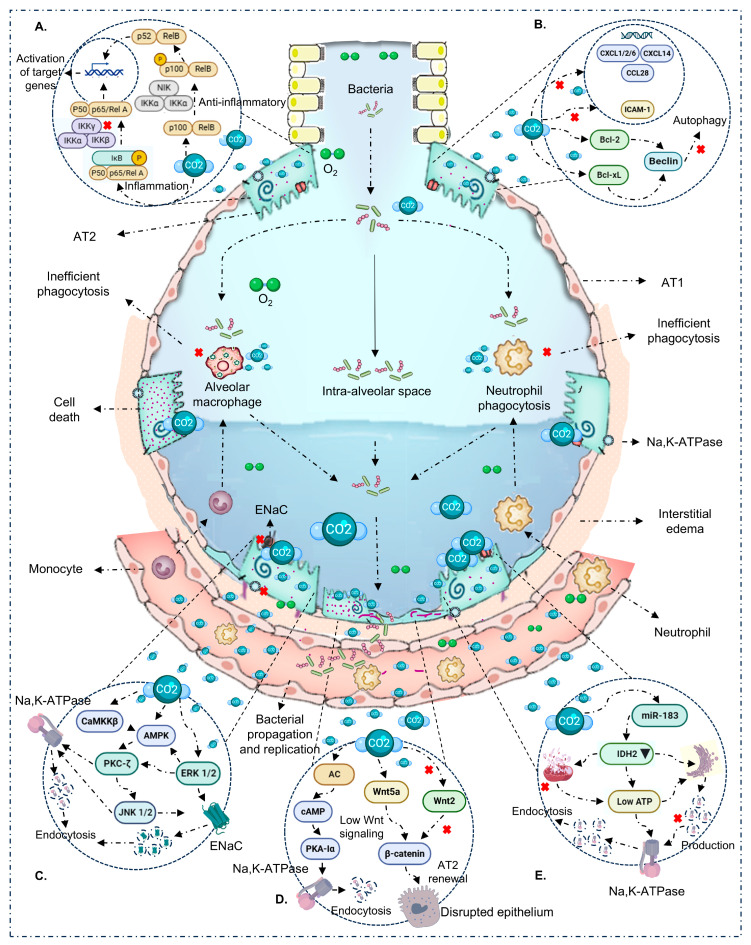
Immuno-epithelial mechanisms modulated by hypercapnia/hypercapnic acidosis. (**A**) NF-κB: CO_2_ dampens the canonical arm (↓ IKK/p65, ↑ IκBα) and can activate the non-canonical arm (p100 → p52/RelB), reprogramming inflammatory output. (**B**) Autophagy: CO_2_ promotes Bcl-2/Bcl-xL binding to Beclin-1, inhibiting class III PI3K and autophagosome initiation; this associates with inefficient phagocytosis/bacterial clearance during infection. (**C**) Ion transport: activation of AMPK/PKA/CaMKKβ/PKC-ζ/ERK1/2 promotes ENaC and Na,K-ATPase endocytosis, reducing AFC. (**D**) Parallel routes: (**i**) adenylyl cyclase raises cAMP, phosphorylating PKA-Iα and driving Na,K-ATPase endocytosis/barrier disruption; (**ii**) depressed Wnt/β-catenin signaling (↓ β-cat, ↓ AT2 proliferation) compromises resealing/repair. (**E**) Mitochondrial energy/miR-183: ATP/IDH2 alterations and miR-183 favor Na,K-ATPase endocytosis and epithelial dysfunction. With prolonged exposure and/or high PaCO_2,_ these routes converge on barrier disruption and bacterial spread; with brief, moderate exposure without infection, protective effects may prevail.

**Table 1 ijms-26-09622-t001:** General characteristics of included studies (chronological order by year of publication).

Citation	Year	Country	Type of Study
Vadász I [26]	2008 Jan	USA	Experimental
O’Croinin DF [21]	2008 Jul	Ireland	Experimental
Liu Y [27]	2008 Oct	USA	Experimental
Chonghaile MN [10]	2008 Nov	Ireland	Experimental
Ni Chonghaile M [28]	2008 Dec	Ireland	Experimental
Abolhassani M [29]	2009 Apr	France	Experimental
Nichol AD [22]	2009 Nov	Ireland	Experimental
O’Toole D [30]	2009 Nov	Ireland	Experimental
Higgins BD [31]	2009 Dec	Ireland	Experimental
Wang N [31]	2010 Jul	USA	Experimental
Welch LC [18]	2010 Sep	USA	Experimental
Peltekova V [32]	2010 May	Canada	Experimental
Cummins EP [33]	2010 Oct	Ireland	Experimental
Vohwinkel CU [34]	2011 Oct	USA	Experimental
Oliver KM [17]	2012 Apr	Ireland	Experimental
Contreras M [20]	2012 Sep	Ireland	Experimental
Vadász I [35]	2012 Oct	USA	Experimental
Lecuona E [19]	2013 May	USA	Experimental
Yang WC [36]	2013 Oct	China	Experimental
Gates KL [23]	2013 Nov	USA	Experimental
Nardelli LM [37]	2015 Jan	Brazil	Experimental
Casalino-Matsuda SM [25]	2015 Jun	USA	Experimental
Dada L [38]	2015 Dec	USA	Experimental
Yang W [39]	2015 Dec	China	Experimental
Masterson C [40]	2016 Apr	Ireland	Experimental
Horie S [41]	2016 Dec	Ireland	Experimental
Gwoździńska P [42]	2017 May	Germany	Experimental
Keogh CE [16]	2017 Jul	Ireland	Experimental
Casalino-Matsuda SM [43]	2018 Sep	USA	Experimental
Cortes-Puentes GA [44]	2019 Jan	USA	Experimental
Kryvenko V [45]	2020 Feb	Germany	Experimental
Casalino-Matsuda SM [46]	2021 Apr	USA	Experimental
Gabrielli NM [47]	2021 Jul	Germany	Experimental
Kryvenko V [48]	2021 Dec	Germany	Experimental
Dada L [49]	2023 Feb	Germany	Experimental

**Table 2 ijms-26-09622-t002:** Pathway-level results (canonical/non-canonical NF-κB, AMPK/PKA), innate immunity outputs (cytokines, phagocytosis, autophagy), and epithelial outcomes (ENaC, Na,K-ATPase, fluid clearance), with effect direction and mechanistic notes per study.

Author	Model	Context	CO_2_ Exposure (Metric; Duration; pH)	Primary Finding
NF-κB signaling (canonical and non-canonical)				
Liu Y et al., 2008 [27]	In vitro: human pulmonary microvascular endothelial cells	Non-infectious (LPS/TNF-α; infection-mimic)	FiCO_2_ 5% and 10%; 4 h; unbuffered metabolic acidosis	Harmful: NF-κB (canonical) expression ↑ after 4 h hypercapnia + acidosis; pro-inflammatory signal
Abolhassani M et al., 2009 [29]	In vivo (rat) + in vitro: HT-29 (human colon); A549 (alveolar type II epithelium)	Non-infectious	FiCO_2_ 5%, 10%, 15%; 1 h (in vivo); pH n.s	Harmful: PP2A activity ↑ with p65 (NF-κB) nuclear translocation ↑ (pro-inflammatory signal)
O’Toole D et al., 2009 [50]	In vitro: human bronchial epithelial (primary), primary small airway epithelial cells, and A549 (alveolar type II)	Non-infectious	FiCO_2_ 5%, 10%, 15%; 24 h; unbuffered metabolic acidosis	Harmful: Hypercapnia inhibits canonical NF-κB and delays epithelial repair; confirmed with IκBα transgene (super-repressor)
Wang N et al., 2010 [31]	In vitro: THP-1 (human), human alveolar macrophages, RAW 264.7 (mouse)	Non-infectious (LPS/TLR stimulation; infection-mimic)	FiCO_2_ 5%, 9%, 12.5%, 20%; 6 h; pH-independent	Harmful: TNF and IL-6 mRNA induction ↓ under hypercapnia; IκBα and RelA/p65 phosphorylation unaffected → immunosuppressive signature (canonical NF-κB not the driver)
Cummins EP et al., 2010 [33]	In vitro: mouse embryonic fibroblasts; A549 (alveolar epithelium); primary human cells	Infectious (LPS stimulation)	FiCO_2_ 5%, 10%; 4 h; Buffered (pH 7.4)	Harmful: Hypercapnia blocks IκBα phosphorylation/degradation and p65 nuclear translocation, thereby inactivating canonical NF-κB → immunosuppressive signature
Oliver KM et al., 2012 [17]	In vivo (rats) and in vitro: alveolar epithelial A549 cells and mouse embryonic fibroblast	Non-infectious (LPS/TLR stimulation; infection-mimic)	FiCO_2_ 5%, 10%; 1.5 h; pH-independent	Protective: Hypercapnia promotes p100→p52 cleavage and RelB nuclear localization (non-canonical NF-κB) → anti-inflammatory/immunosuppressive effect
Contreras M et al., 2012 [20]	In vivo (rat) + in vitro: A549 (alveolar epithelium)	Non-infectious (mechanical stretch/VILI-like)	FiCO_2_ 5%, 8%, 10%, 15%, 20%; 4 h; unbuffered metabolic acidosis	Protective: Hypercapnia inactivates canonical NF-κB in vivo/in vitro, preserves cytoplasmic IκBα, and ↓ IL-8 → anti-inflammatory in a sterile context
Yang W et al., 2015 [39]	In vivo (rats) + in vitro (unspecified cell models)	Non-infectious (CO_2_ ventilation; no injurious stimulus)	PaCO_2_ 35–150 mmHg; 4 h; unbuffered metabolic acidosis	Protective: Hypercapnia maintains cytoplasmic IκBα and reduces canonical NF-κB activity → protective signal in a sterile context
Masterson C et al., 2016 [40]	**In vivo (rats)** + **in vitro**: small airway epithelial (SAE), human bronchial epithelial, A549 (alveolar)	Infectious (*Escherichia coli*; 4 h challenge)	FiCO_2_ 5%, 10%, 15%; 4 h; unbuffered metabolic acidosis	Protective: p65 nuclear translocation ↓; IκBβ phosphorylation ↓; IκBα preserved/↑; NF-κB remains inactivated → anti-inflammatory signature
Horie S et al., 2016 [41]	In vitro: bronchial and alveolar A549 cells	Non-infectious (mechanical lung stretch)	FiCO_2_ 5%, 12%; duration: 24 h; unbuffered metabolic acidosis	Protective: IκBα phosphorylation ↓; IκBα degradation ↓; NF-κB activity ↓; IL-8 ↓
Keogh CE et al., 2017 [16]	In vitro: alveolar epithelial A549, human embryonic kidney (HEK), mouse embryonic fibroblasts (MEFs)	Non-infectious (no LPS/inflammatory stimulation; infection-mimic)	FiCO_2_ 5%, 8%; 1.25 h; pH: n.s.	Protective: Non-canonical NF-κB (p100→p52/RelB) ↑; RelB nuclear localization ↑ → anti-inflammatory/immunosuppressive signature
Stress-kinase signaling (ASK1/JNK/p38)				
Yang WC et al., 2013 [36]	In vivo (rats) + in vitro: alveolar type II epithelial cells (AT2)	Non-infectious (high-pressure ventilation; VILI-like)	PaCO_2_ 80–100 mmHg; 4 h; unbuffered metabolic acidosis	Protective: Hypercapnia decreases ASK1/JNK and p38 MAPK activities → protective signal (↓ vascular leak/oxidative stress)
Innate immunity outputs (cytokines, phagocytosis, autophagy)				
O’Croinin DF et al., 2008 [21]	In vivo (rats)	Infectious (pneumonia, no antibiotics)	Inspired CO_2_: 5%; 48 h; unbuffered metabolic acidosis	Harmful: Neutrophil phagocytosis ↓; ALI severity ↑ → worse infection control
Liu Y et al., 2008 [27]	In vivo (rabbits)	Non-infectious (LPS/TNF-α; infection-mimic)	FiCO_2_ 5%, 10%; 4 h; unbuffered metabolic acidosis	Harmful: Pro-inflammatory outputs ↑ (IL-8 ↑; VCAM-1 ↑; E-selectin ↑; P-selectin ↑)
Chonghaile MN et al., 2008 [10]	In vivo (rats): established pneumonia; antibiotics	Infectious (*Escherichia coli*)	FiCO_2_ 5%; 6 h; unbuffered metabolic acidosis	Protective: BAL TNF-α ↔; IL-6 ↔; BAL neutrophils ↔ (±antibiotics); with antibiotics: bacterial count ↓; histological lung injury ↓ → beneficial with antibiotics; neutral without
Ni Chonghaile M et al., 2008 [28]	In vivo (rats): established pneumonia, antibiotics, no antibiotics	Infectious (*Escherichia coli*)	FiCO_2_ 5%; 6 h; unbuffered metabolic acidosis	Protective: Lung injury ↓; BAL neutrophils unchanged; BAL TNF-α/IL-6 unchanged; lung bacterial load unchanged → protective signature, apparently neutrophil-independent
Abolhassani M et al., 2009 [29]	In vivo (rat) + in vitro: HT-29 (human colon); A549 (alveolar type II epithelium)	Non-infectious	FiCO_2_ 5%, 10%, 15%; 1 h (in vivo); pH n.s	Harmful: Pro-inflammatory gene expression ↑ (RANTES, MIP-1α, MIP-1β, MCP-1, TCA-3, IP-10, IL-6, IL-8); MUC5AC ↑; airway hyperreactivity ↑ → harmful/pro-inflammatory signature.
Nichol AD et al., 2009 [22]	In vivo (rats)	Infectious	PaCO_2_ 64–80 mmHg; 6 h; buffered	Harmful: IL-1β ↑; BAL neutrophils ↑; lung structural damage ↑ → harmful/pro-inflammatory despite normal pH
Higgins B et al., 2009 [30]	In vivo (rats)	Infectious—systemic sepsis (**cecal** ligation and puncture)	FiCO_2_ 5%, 8%; 96 h; two conditions: BHC (buffered/normalized pH) vs. HCA (unbuffered hypercapnic acidosis)	Harmful: BHC: BAL IL-6 ↓; BAL neutrophils ↔; BAL TNF-α ↔; bacterial load ↔; neutrophil phagocytic function ↔ Protective: BAL TNF-α ↓; BAL IL-6 ↔; lung histologic injury ↓; rate of bacteremia entry ↓
Wang N et al., 2010 [31]	In vitro: THP-1 (human), human alveolar macrophages, RAW 264.7 (mouse)	Non-infectious (LPS/TLR stimulation; infection-mimic)	FiCO_2_ 5%, 9%, 12.5%, 20%; 6 h; pH-independent	Harmful: Macrophage phagocytosis ↓ under hypercapnia → impaired bacterial clearance/immunosuppressive signature
Peltekova V et al., 2010 [32]	In vivo (rats)	Non-infectious—ventilator-induced lung injury (VILI)	FiCO_2_ 0%, 5%, 12%, 25%; 3 h; unbuffered metabolic acidosis	Protective: IL-6 ↓; KC ↓; MCP-1 ↓; TNF-α ↓; elastance rise attenuated; microvascular leak ↓; histology improved; MPO+ cells ↓; COX-2 (mRNA/protein) ↓; eicosanoids ↓ (modest); tissue nitrotyrosine ↑ → net protective signal on injury/innate inflammation with caveat (nitrotyrosine)
Cummins EP et al., 2010 [33]	In vitro: mouse embryonic fibroblasts; A549 (alveolar epithelium); primary human cells	Infectious (LPS stimulation)	FiCO_2_ 5%, 10%; 4 h; buffered (pH 7.4)	Harmful: CCL2/MCP-1 ↓; ICAM-1 ↓; TNF-α ↓; IL-10 ↑ → anti-inflammatory/immunosuppressive
Oliver KM et al., 2012 [17]	In vivo (rats) and in vitro: alveolar epithelial A549 cells and mouse embryonic fibroblast	Non-infectious (LPS/TLR stimulation; infection-mimic)	FiCO_2_ 5%, 10%; 1.5 h; pH-independent	Protective: TNF-α mRNA ↓; COX-2 ↓ under elevated CO_2_ (pH-independent) → anti-inflammatory/immunosuppressive signature
Contreras M et al., 2012 [20]	In vivo (rat) + in vitro: A549 (alveolar epithelium)	Non-infectious (mechanical stretch/VILI-like)	FiCO_2_ 5%, 8%, 10%, 15%, 20%; 4 h; unbuffered metabolic acidosis	Protective: PaO_2_ ↑; lung compliance ↑; BAL protein ↓; BAL neutrophils ↓; BAL IL-6/TNF-α/CINC-1 ↓ → anti-inflammatory, tissue-protective signal in sterile VILI
Yang WC et al., 2013 [36]	In vivo (rats) + in vitro: alveolar type II epithelial cells (AT2)	Non-infectious (high-pressure ventilation; VILI-like)	PaCO_2_ 80–100 mmHg; 4 h; unbuffered metabolic acidosis	Protective: BAL (TNF-α ↓, MIP-2 ↓, neutrophil recruitment ↓); oxidative stress/injury: MDA ↓, SOD ↑, MPO ↓, LDH ↓; apoptosis: cleaved caspase-3 ↓, early/late apoptosis ↓ → protective signal (vascular leak/oxidative stress ↓)
Gates KL et al., 2013 [23]	In vivo (rats)	Infectious (pneumonia, no antibiotics)	Inspired CO_2_: 5%, 10%; 96 h; unbuffered metabolic acidosis	Harmful: Neutrophil phagocytic capacity ↓; bacterial load ↑; dissemination to other organs ↑; early cytokines (IL-6, TNF) ↓ → immunosuppressive signature with worse infection control
Nardelli LM et al., 2015 [37]	In vivo (rats)	Non-infectious (Paraquat)	PaCO_2_ ventilation: 35–80 mmHg	Hypercapnia, independent of acidosis, reduces IL-6, IL-1β, and type III pro-collagen expression. It also decreases neutrophil count and apoptosis processes
Casalino-Matsuda SM et al., 2015 [25]	In vitro: human alveolar macrophages; THP-1 (human monocytic leukemia); HeLa GFP-LC3	Infectious	FiCO_2_ 5%, 15% CO_2_; 18 h; pH n.s	Harmful: Bcl-2/Bcl-xL ↑ → Beclin-1 sequestration ↑ → autophagosome initiation ↓ /autophagic flux ↓; bacterial killing ↓ → immunosuppressive signature
Yang W et al., 2015 [39]	In vivo (rats) + in vitro (unspecified cell models)	Non-infectious (CO_2_ ventilation; no injurious stimulus)	PaCO_2_ 35–150 mmHg; 4 h; unbuffered metabolic acidosis	Protective: BAL neutrophils ↓; total BAL cells ↓; MPO ↓; TNF-α ↓; IL-1β ↓; MIP-2 ↓ → protective anti-inflammatory signal in a sterile context
Casalino-Matsuda SM et al., 2018 [43]	In vitro: human bronchial epithelial cells	Non-infectious	FiCO_2_ 20%; 24 h; pH: n.s.	Harmful. Immunoregulatory gene program Δ (CXCL1, CXCL2, CXCL14, CCL28, IL-6R, TLR4 altered under sustained hypercapnia; direction context-dependent)
Casalino-Matsuda SM et al., 2021 [46]	In vitro: human monocytic leukemia THP-1 and mouse monocyte–macrophage RAW 264.7	Non-infectious (PAMP stimulation)	5%, 20%; 3 h; unbuffered metabolic acidosis	Harmful: LPS-upregulated innate/antiviral/type-I IFN programs ↓ (NF-κB1/2, REL/RELB, STAT1/2, IRF1/7, DDX58, IL6, CCL2, ICAM1 ↓) → net immunosuppressive signature
cAMP/PKA–AMPK pathways and epithelial transport (ENaC; Na,K-ATPase; PKC-ζ; CaMKKβ)				
Vadász I et al., 2008 [26]	In vivo (rats), ex vivo (perfused rat lung), and in vitro (primary rat AECII and human A549 cells)	Non-infectious	PaCO_2_: 60–120 mmHg; 24 h; unbuffered metabolic acidosis	Harmful: AFC ↓; [Ca^2+^]i ↑ → CaMKKβ ↑ → AMPK (Thr172) ↑ → PKC-ζ translocation/activity ↑ → Na,K-ATPase endocytosis ↑ → epithelial transport failure, edema resolution worsens
Liu Y et al., 2008 [27]	In vivo (rabbits)	Non-infectious (sterile stimulation with LPS/TNF-α)	FiCO_2_ 5% and 10%; 4 h; unbuffered metabolic acidosis	Alveolar transudation ↑; septal edema ↑; vascular/extravascular leak ↑; alveolar structural damage ↑ → harmful epithelial/Barrier outcome
Nichol AD et al., 2009 [22]	In vivo (rats)	Infectious	PaCO_2_ 64–80 mmHg; 6 h; buffered	Harmful: lung structural damage ↑; lung cell wound repair rate ↓ → harmful/pro-inflammatory despite normal pH
O’Toole D et al., 2009 [50]	In vitro: human bronchial epithelial (primary), primary small airway epithelial cells, and A549 (alveolar type II)	Non-infectious	FiCO_2_ 5%, 10%, 15%; 24 h; unbuffered metabolic acidosis	Harmful: hypercapnia inhibits canonical NF-κB and delays epithelial repair
Welch LC et al., 2010 [18]	In vivo (rats) and In vitro: ATII (rats), A549 (alveolar type II) y A549–GFP–α1	Non-infectious	FiCO_2_ 5%, 20%; 0.25 h; buffered	Harmful: ERK1/2 ↑ (minutes) → AMPK ↑ → Na,K-ATPase endocytosis ↑ → AFC ↓ → epithelial transport failure
Vohwinkel CU et al., 2011 [34]	In vitro: A549 (alveolar type II) and fibroblasts	Non-infectious	FiCO_2_ 5%, 7%; 72 h; buffered	Harmful: miR-183 ↑ → IDH2 (mRNA/protein) ↓ → TCA flux ↓ → mitochondrial dysfunction ↑; epithelial proliferation/repair ↓ → harmful metabolic/repair signature
Vadász I et al., 2012 [35]	In vitro: Alveolar epithelial cells	Non-infectious	PaCO_2_ 60–120 mmHg; 1 h; pH: n.s.	Harmful: AMPK → PKC-ζ → JNK(Ser129)↑ → endocitosis de Na,K-ATPase ↑ → AFR ↓; pH-independent → epithelial transport failure
Lecuona E et al., 2013 [19]	In vitro: A549 (alveolar type II), rat RLE-6TN cells, and primary rat alveolar epithelial type II	Non-infectious	PaCO_2_ 40–120 mmHg; 0.5 h; buffered	Harmful: sAC–cAMP microdomains ↑ → PKA-RIα ↑ → α-adducin Ser726 phosphorylation ↑ → Na,K-ATPase endocytosis ↑ → AFC ↓ → epithelial transport failure
Dada L et al., 2015 [38]	In vitro: A549 (alveolar type II) and rat type II cells	Non-infectious	FiCO_2_ 5%, 10%, 15%, 20%; 0.5 h; buffered	Harmful: AMPK ↑ → JNK ↑ → LMO7b Ser1295-P ↑ → LMO7b–Na,K-ATPase interaction ↑ → clathrin/AP2 recruitment ↑ → Na,K-ATPase endocytosis ↑ → AFC ↓; pH-independent → epithelial transport failure
Gwoździńska P et al., 2017 [42]	In vitro: A549 (alveolar type II)	Non-infectious	FiCO_2_ 5%; 0.5 h; buffered	Harmful: ERK1/2 ↑; AMPK-α1 ↑; JNK1/2 ↑; Nedd4-2 pT899 ↑; β-ENaC pT615 ↑; β-ENaC poly-Ub ↑; ENaC endocytosis ↑; α-ENaC Ub ↔ → ENaC surface ↓; epithelial Na^+^ transport/AFC ↓ → epithelial transport failure; pH-independent
Cortes-Puentes et al., 2019 [44]	In vitro: A549 (alveolar type II) and rat alveolar epithelial cell type I	Non-infectious	PaCO_2_ 80 mmHg; 0.25 h; buffered	Harmful: under unbuffered HCA: AC activity ↓ → cAMP ↓ → PKA signaling ↓ → epithelial repair/restitution rate ↓ (harmful). Buffering (pH normalization) rescues AC/cAMP and epithelial repair → pH-dependent
Kryvenko V et al., 2020 [45]	In vitro: A549 (alveolar type II) and rat type II cells	Non-infectious	FiCO_2_ 5%, 10%, 20%; 12 h; pH: n.s.	Harmful: ER oxidation ↑ → misfolded Na,K-ATPase β retained in ER ↑ → α:β assembly ↓ → plasma-membrane Na,K-ATPase ↓ /pump activity ↓; calnexin/BiP association ↑; mitochondrial/ATP deficit component ↑ → AFC ↓→ epithelial transport failure
Gabrielli et al., 2021 [47]	In vitro: A549 (alveolar type II) and primary rat alveolar epithelial type II	Non-infectious	FiCO_2_ 5%, 20%; 0.5 h; buffered	Harmful: PKC-ζ-dependent β-Ser11 phosphorylation ↑ → TRAF2 (E3) recruitment ↑ → Na,K-ATPase β polyubiquitination (K5/K7) ↑ → endocytosis ↑ + proteasomal degradation ↑ → PM Na,K-ATPase ↓ → AFC ↓ → epithelial transport failure
Kryvenko V et al., 2021 [48]	In vitro: A549 (alveolar type II) and primary rat alveolar epithelial type II	Non-infectious	FiCO_2_ 5%, 7%, 10%, 20%; 1 h; buffered	Harmful: IP_3_R–Ca^2+^ release ↑ → IRE1α pSer724 ↑ → MAN1B1/EDEM1–ERAD ↑ → proteasomal degradation ↑ → Na,K-ATPase β in ER ↑ (retention)/at plasma membrane ↓ → AFC ↓→ epithelial transport failure
Dada L et al., 2023 [49]	In vivo (rats) and in vitro: A549 (alveolar type II), and primary rat alveolar epithelial type II	Non-infectious	FiCO_2_ 5%, 10%, 20% 24 h/504 h; pH: n.s.	Harmful: Wnt5a ↑; Wnt2 ↓; β-catenin signaling (Axin2) ↓ in AT2 → AT2 proliferation/repair ↓ → alveolar repair suppressed

Abbreviations: ACEII: alveolar epithelial type II cells; AFC: alveolar fluid clearance; AMPK: AMP-activated Protein Kinase; ASK1: Apoptosis Signal-Regulating Kinase 1; AT2: alveolar type II; BAL: Bronchoalveolar Lavage; Bcl-2: B-cell Lymphoma 2; Bcl-xL: B-cell Lymphoma-extra-large Protein; cAMP: Cyclic Adenosine Monophosphate; BHC: Buffered Hypercapnia; CCL2: Chemokine Ligand 2; CXCL1: Chemokine Ligand 1; CXCL2: Chemokine Ligand 2; CXCL14: Chemokine Ligand 14; CCL28: Chemokine Ligand 28; ENaC: Epithelial Sodium Channels; ERAD: Endoplasmic Reticulum-Associated Degradation; ERK1/2: Extracellular Signal-Regulated Kinases 1/2; HCA: hypercapnic acidosis; ICAM-1: Intercellular Adhesion Molecule 1; IDH2: Isocitrate Dehydrogenase 2; IκB: Inhibitor of kappa B; IL-1: Interleukin 1; IL-6: Interleukin 6; IL-8: Interleukin 8; IP_3_R: Inositol 1,4,5-Trisphosphate Receptor; IRE1α: Inositol-Requiring Enzyme 1 alpha; JNK: c-Jun N-terminal Kinase; LDH: Lactate Dehydrogenase; LPS: Bacterial Lipopolysaccharide; MDA: Malondialdehyde; MPO: Myeloperoxidase; mRNA: Messenger Ribonucleic Acid; NF-κB: Nuclear Factor-kappa B; PAMP: Pathogen-associated Molecular Patterns; PKA: Protein Kinase A; PKA: Protein Kinase A Regulatory Subunit I alpha; PKC -ζ: Protein Kinase C zeta; PM: plasma membrane; PP2A: Protein Phosphatase 2A; poly-Ub/Ub: polyubiquitination/ubiquitin; p38 MAPK: p38 mitogen-activated protein kinase; TLR: Toll-Like Receptor; TLR4: Toll-Like Receptor 4; sAC: Soluble Adenylyl Cyclase; SOD: Superoxide Dismutase; TNF: Tumor Necrosis Factor; VCAM-1: Vascular Cell Adhesion Molecule-1. Conventions: ↑ increase/activation; ↓ decrease/inhibition; ↔ no change, n.s. = not significant.

## Data Availability

All data, extraction templates, decision log, PRISMA-ScR checklist, and full search strategies are openly available at OSF (https://doi.org/10.17605/OSF.IO/WV85T).

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
