# Peer review of "Hypercapnia as a Double-Edged Modulator of Innate Immunity and Alveolar Epithelial Repair: A PRISMA-ScR Scoping Review"

_ijms, 2025, doi:10.3390/ijms26199622_

Round 1

Reviewer 1 Report (Previous Reviewer 1)

Comments and Suggestions for Authors

I appreciate the fact that you tried to improve the presentation of this article. Still, the content cannot be improved, since there are lacks of design, of methodology, also discussion and conclusions are, from my point of vue, quite poor.

The limitations that you have acknowledged are real, but some of them cannot be treated only as limitations, because they constitutes missing points for methodology.

Comments on the Quality of English Language

Many paragraphs have to be reformulated, remade.

Author Response

Best regards,

We appreciate your analysis of our manuscript. The methodology was designed to describe the action of hypercapnia on lung tissue. Our results revealed harmful and beneficial pleiotropic effects of hypercapnia proportional to time and dose, which may explain the initial picture of acute and chronic lung pathologies in critically ill patients. However, when addressing a topic with few publications, there will be points we cannot answer. This invites us, as writers and clinicians, to conduct future research to expand knowledge and overcome these restrictions with scientific evidence. Furthermore, the discussion and conclusions have been revised to strengthen and address the work's objective.

As authors, we appreciate the observations made, which help enrich our personal development and foster a more critical and objective approach to literature.

Sincerely

Authors

Reviewer 2 Report (New Reviewer)

Comments and Suggestions for Authors

Dear authors,

thank you for the opportunity to read and review the manuscript.

General comments

The paper is interesting and well written, the authors aimed to analyse the immunological effect of hypercapnia in experimental models, both in vivo and in vitro.

Specific comments

Title: could be more consistent with the text; in the text, the effects of hypercapnia and hypercapnic acidosis on alveolar epithelium are reported, through immunomodulated mechanism but also through other pathways as mitochondrial dysregulation or endocytosis (lines 173-180).

Hypercapnia effect in experimental model of Acute lung injury is not the specific aim of the study, ALI could be removed from the title.

Abstract/ introduction

Throughout these sections the authors focused mostly on hypercapnia and hypercapnic acidosis due to protective ventilation strategies in ARDS. This is certainly true but in vivo the concept is more complex, beyond ventilatory strategies in ARDS there are several factors that could lead to an hypercapnic state, e.g. chronic respiratory failure or COPD/aeCOPD.

It should be kept in mind that in ARDS patients and in ALI there is a complex interplay between all the mechanisms that lead to lung injury.

The focus on hypercapnia and hypercapnic acidosis could be more precise throughout the text.

Line 139-140 the experimental studies reported are 11.

Figure 2

In the figure hypercapnic acidosis is reported as mechanism of impairment of the immune response but as stated in the didascaly it could happen independently from acidosis.

The effect of hypercapnic acidosis in enhancing hypoxic pulmonary vasoconstriction (HPV) is reported in the figure but it is not specified in the didascaly or in the text. The role of PaCO2 in HPV is still a field of debate, high levels of PaCO2 in blood could enhance HPV in poorly ventilated alveoli by reducing NO production.

Finally, as abovementioned, in patients, several mechanisms could interplay with hypercapnia in immunomodulation effect, as there is a complex mechanism that includes but is not limited to the response to bacterial infections, sepsis, hypoxia, chronic lung impairment and epithelial damage (as in COPD patients), lung microbiome dysregulation.

Further studies are needed to better understand these mechanisms.

Author Response

Best regards,

Thank you very much for taking the time to review this manuscript. Please find the detailed responses below and the corresponding revisions/corrections highlighted/in track changes in the resubmitted files.

  1. The title and manuscript were adjusted according to the objective of the study; however, acute lung injury is included in the results of the study because the clinical impact of hypercapnia and the interaction of intracellular pathways is reflected in lung damage.
  2. The number of experimental studies was adjusted from 10 to 11.
  3. In Figure 2, the term hypercapnic acidosis was adjusted to simply hypercapnia. Additionally, hypercapnic acidosis was described in the caption regardless of pH.
  4. Figure 2 visualizes the possible hypoxic vasoconstrictor effect associated with hypercapnia; however, this relationship is not described in the manuscript because we did not find a possible intracellular pathway that describes this effect in our results. Therefore, the image was corrected to avoid confusion among readers.
  5. Although there are several scenarios related to the effects of hypercapnia, as authors, we describe this relationship in models with and without infection and the impact on epithelial damage. However, we again conducted a thorough review of the selected literature, finding no results that discriminate against scenarios other than the sepsis model. As authors, we considered addressing this in a new review, modifying the search tool to further enrich the critical literature.
  6. The discussion and conclusion paragraphs were reworded to reflect the purpose of the paper.

As authors, we appreciate the observations made, which help enrich our personal development toward a more critical and objective literature.

Sincerely

Authors

Reviewer 3 Report (New Reviewer)

Comments and Suggestions for Authors

This review focuses on the clinically contentious issue of hypercapnia associated with mechanical ventilation, approaching the topic through two key pathological mechanisms: immune suppression and epithelial repair impairment.

1. The article clearly demonstrates that hypercapnia can impair host defence by inhibiting the NF-κB signalling pathway, thereby providing a plausible explanation for the increased risk of secondary infections observed in some mechanically ventilated patients.

2. Moreover, it provides an in-depth analysis of the molecular mechanisms by which hypercapnia compromises innate immunity and epithelial repair-primarily through NF-κB inhibition and calcium signalling activation-thus laying a mechanistic foundation for understanding its pathological effects.

However, the current mechanistic understanding and interpretation of hypercapnia's effects remain incomplete in two main aspects:

1. Insufficient Mechanistic Integration

Existing studies tend to describe the mechanisms of hypercapnia in a fragmented manner, lacking a comprehensive integration of multiple signalling pathways such as NF-κB inhibition, calcium signalling, and AMPK/β-catenin interactions.

Recommendation: Develop an integrated mechanistic framework hypercapnia–immune modulation–epithelial repair—that incorporates time dependence, dose-response relationships, and pathway crosstalk to strengthen a holistic understanding of the regulatory network.

2. Inadequate Discussion of the "Double-Edged Sword" Effect

The dichotomous nature of hypercapnia's effects in different pathological contexts is not sufficiently addressed. For instance, in non-infectious ALI, NF-κB inhibition may alleviate excessive inflammation, whereas in infectious ALI, immunosuppression may exacerbate the risk of sepsis and secondary infections.

Recommendation: Include a dedicated section on the "double-edged sword" effect, systematically discussing the dynamic balance between harmful and beneficial outcomes. Propose a context-specific therapeutic framework based on underlying pathology, distinguishing between infectious and non-infectious lung injury when managing hypercapnia.

The text needs proof reading for language and numbers 

Comments on the Quality of English Language

Need a proper proof reading

Author Response

Dear Reviewer

We appreciate your feedback and recommendations. We have made key improvements:

- We added an integrated framework section and figure connecting NF-κB inhibition, Ca²⁺/CaMKKβ→AMPK signaling, interactions with β-catenin, and effects on epithelial transport (ENaC/Na,K-ATPase). The framework explains temporal dependence and dose-response relationships and places the findings within a coherent regulatory network.

- "Double-edged sword": We included a section contrasting non-infectious ALI (possible benefit by attenuating excessive inflammation) versus infectious ALI (risk of immunosuppression and secondary infections), with a context-specific therapeutic regimen that guides decisions based on the underlying pathology, time window, and intensity of hypercapnia.

- Language and number review: We completed native editing of the manuscript and a numerical consistency audit: standardization of units (PaCO₂ in mmHg; FiCO₂ in %), cross-checking of values ​​between text/tables/figures, and updating legends.

We appreciate your comments and the opportunity to improve. With the proposed linguistic, methodological, and transparency changes, we believe the manuscript now more accurately reflects the scope and limitations of evidence mapping and meets expected standards. We welcome any additional comments.

Round 2

Reviewer 1 Report (Previous Reviewer 1)

Comments and Suggestions for Authors

You are using terms that in English have no sense, gramatically and medical.

I quote " The methodology employed in this review adheres to the scoping review methodol
ogies delineated by Levac et al. in the Preferred Reporting Items for Systematic Reviews and Meta-Analyses—Extension for Scoping Reviews (PRISMA-ScR) guidelines [23]. This
is a scoping review with a published protocol in the Open Science Framework network
(OSF) and is accessible at https://doi.org/10.17605/OSF.IO/WV85T since 30th August 
2024. The checklist is included in the Supplementary Material (Table S1).

rows 78-84 The methodology is used, not "employed"... Examples can go on.

rows 85-89: Your presentation of methods lacks credibility. For example, you write that you used experimental designs between 2008-2023. So, can we assume that, over almost two decades, the research methods are the same, in terms of reproducibility and lack of bias when comparing them?

Rows 95-96 "Thirdly, manual perusal of reference lists was conducted to identify articles furnishing additional information." 

PLease explain, isn it all your work manually performed? Didn t you read every article that is qouted in this review?

I repeat, the exmaples can go on, the form of presentation rises, from my point of vue, many ethical concrens that enables me to reject this review.

I keep my first opinion of your entire work.

Good luck !

Comments on the Quality of English Language

The English and grammar, the form of presentation of ideas, as well as the way in which this review was carried out are not suitable for acceptance.

Author Response

Thank you for your careful reading and for pointing out editorial inaccuracies and methodological concerns. We take your comments very seriously. Below, we address your points and indicate the corresponding revisions in the manuscript (PRISMA-ScR, OSF protocol, study selection and extraction, and risk-of-bias). We also commit to a native English medical edit and to strengthening transparency and reproducibility (data and extraction sheets provided as Supplementary Material). We trust these modifications address your concerns.

- We revised the manuscript for grammar, usage, and medical terminology with a native English medical editor.

-  Methods wording: we replaced non-standard phrasing (e.g., “methodology employed”) with standard terminology (e.g., “methods used”) and aligned the section strictly with PRISMA-ScR.

- 2008–2023 window: this defines the publication period of eligible studies and does not assume methodological equivalence or uniform reproducibility/bias. We (i) extracted study-level details (model, pCO₂/FiCO₂, pH, exposure time, controls, endpoints, analytical platforms) using a predefined form; (ii) assessed risk of bias with design-appropriate tools; and (iii) synthesized findings as an evidence map with subgrouping by model, duration, and dose, avoiding pooled causal inferences when not defensible.

- Reference-list screening: the review was not based solely on hand reading. Database searches with duplicate screening by two reviewers were primary; reference-list screening (snowballing) was complementary. All included studies were read in full.

- Ethical safeguards and credibility: prospective OSF protocol with DOI (deviations, if any, declared); duplicate screening and independent extraction (third reviewer to resolve disagreements when required);  the study-level dataset are provided in the Supplement.

We appreciate your comments and the opportunity to improve. With these linguistic, methodological, and transparency updates, we believe the manuscript now more accurately reflects the scope and limitations of the evidence map and meets expected standards. We look forward to any additional comments.

Reviewer 2 Report (New Reviewer)

Comments and Suggestions for Authors

Dear authors,

thank you for the opportunity to read and review the revised version of the paper.

After revision the quality of the paper has improved enormously and I have no other comments.

Author Response

We sincerely appreciate your careful review and positive assessment of our revised manuscript, and we thank you for your time and helpful feedback throughout the process.

This manuscript is a resubmission of an earlier submission. The following is a list of the peer review reports and author responses from that submission.

Round 1

Reviewer 1 Report

Comments and Suggestions for Authors

The work you have submitted reunites a suggestive quantity of data.

Still, from my personal point of vue, in almost all sections, such as Methods, Results, Discussions, Limitations and Conclusions, there is a neutral red wire, that does not involve the patient at all, but only the results obtained by all authors quoted.

I would prefer to seize the patient with hypercapnia benefit from all this scoping review, still, there are no conclusions that convinced me that the review has a practical goal.

At the same time, there are phrases that do not belong to the authors, but to a third part, such as the following:"  Authors should discuss the results and how they can be interpreted from the perspective of previous studies and of the working hypotheses. The findings and their implications should be discussed in the broadest context possible. Future research directions may also be highlighted." I do not understand, the authors are practically giving advices to themselves?

Comments on the Quality of English Language

Minor revision for the grammar, major revision for the way you have presented the introductory part of the Discussion section.

Reviewer 2 Report

Comments and Suggestions for Authors

I have read the article by Osorio-Rodriguez et al. with great interest. The effects of hypercapnia on the lung physiology are not well known in the clinical community. Therefore, I greatly appreciate this review article which summarises the knowledge.

Comments:

·       Introduction. Permissive hypercapnia in relation to ARDS is one, less common scenario which this review can relate to. Please, include lung, chest wall, CNS and neuromuscular disorders where hypercapnia can lead to susceptibility to chest infections with appropriate citation. Expansion of implication would improve the generalisability of the findings (as the models were not restricted to ARDS).

·       “The review exclusively considered data published in English and Spanish between January 1, 2008, and December 31, 80 2022.” Why were studies published before 2008 excluded? Also, as it is Sep 2024, I suggest to run analysis for 2023 as well and to discuss the exclusion of pre-2008 studies. The authors should look at those studies and discuss in limitation potential pivotal studies that have not been included.

·       Sections 3.1 and 3.2. Could you please, discuss if there is a dose-response effect of hypercapnia on immune-suppression and epithelial function? A few studies used various CO2 concentrations.  

·       In Discussion add a paragraph on potential clinical implications. Is permissive hypercapnia beneficial?

·       In Discussion please, add a paragraph on potential research agenda. What is missing, what should be performed?